# Pathophysiology-Based Management of Secondary Injuries and Insults in TBI

**DOI:** 10.3390/biomedicines12030520

**Published:** 2024-02-26

**Authors:** Leonardo de Macedo Filho, Luisa F. Figueredo, Gustavo Adolfo Villegas-Gomez, Matthew Arthur, Maria Camila Pedraza-Ciro, Henrique Martins, Joaquim Kanawati Neto, Gregory J. Hawryluk, Robson Luís Oliveira Amorim

**Affiliations:** 1Neurosurgery Department, Penn State Health Milton S. Hershey Medical Center, Hershey, PA 17033, USA; 2Department of Psychiatry, NYU Langone Health, New York, NY 10016, USA; luisa.figueredoatilano@nyulangone.org (L.F.F.); matthew.arthur@nyulangone.org (M.A.); 3Faculty of Medicine, Universidad de Los Andes, Bogota 111711, Colombia; g.villegasg@uniandes.edu.co; 4Faculty of Medicine, Universidad El Bosque, Bogota 11001, Colombia; mc.pedrazaciromc@gmail.com; 5Postgraduate Program in Health Sciences, Federal University of Amazonas, Manaus 69067-005, Brazil; drhenriquemartinsneuro@gmail.com (H.M.); joaquimkanawati@gmail.com (J.K.N.); amorim.robson@gmail.com (R.L.O.A.); 6Department of Neurosurgery, Cleveland Clinic and Akron General Hospital, Fairlawn, OH 44307, USA; hawrylg@ccf.org; 7Division of Neurological Surgery, Hospital das Clinicas, University of São Paulo School of Medicine, São Paulo 05508-070, Brazil

**Keywords:** secondary TBI, traumatic brain injury, brain injuries, physiopathology

## Abstract

Traumatic Brain Injury (TBI) remains a leading cause of morbidity and mortality among all ages; despite the advances, understanding pathophysiological responses after TBI is still complex, involving multiple mechanisms. Previous reviews have focused on potential targets; however, the research on potential targets has continuously grown in the last five years, bringing even more alternatives and elucidating previous mechanisms. Knowing the key and updated pathophysiology concepts is vital for adequate management and better outcomes. This article reviews the underlying molecular mechanisms, the latest updates, and future directions for pathophysiology-based TBI management.

## 1. Introduction

Traumatic brain injury (TBI) is a leading cause of disability and death among children and young adults, with an incidence of approximately 1.7 million per year in the USA, resulting in 52,000 deaths [1]. Survivors of the initial impact must still contend with the consequences of trauma, as not all injury occurs at the time of impact. The primary injury results from forces applied to the head and involve direct structural damage to the brain. This triggers a cascade of events leading to neurological damage that evolves secondary injury [2]. Several external brain insults, both intracranial and systemic, may complicate and worsen the secondary injury [3]. 

As was pointed out by Ng et al., current therapeutic strategies focus on preventing secondary injury through early surgical intervention, multiparameter monitoring, and targeted treatment in the intensive care setting [2,4]. Adherence to guidelines for managing severe TBI is associated with better outcomes [5]. However, treatment should be individualized to address underlying mechanisms following TBI and the role of secondary injury in recovery. This work focuses on the underlying pathophysiological mechanisms and their related treatments to systematize the thought process for TBI management, highlighting the role. Importantly, although this article focuses on the mechanisms of excitotoxicity, neuroinflammation and oxidative stress, and blood-brain barrier disruption in TBI, there are other mechanisms central to the pathogenesis of TBI, such as changes in energy metabolism and axonal injury. These topics, although critical, are beyond the scope of this review and deserve detailed analysis in future research.

## 2. Materials and Methods

We conducted a literature search of PUBMED, SCOPUS, and Google Scholar from 1980 to 2024 using search terms related to the pathophysiology and treatment of traumatic brain injury on 8 January 2024. The searched keywords were ((Secondary TBI) OR (Secondary Brain Injury) OR (Secondary Traumatic Brain Injury)) AND pathophysiology, retrieving 3043 full-text results. This review (Figure 1) aims to concisely evaluate the pathophysiology of TBI and provide guidance for understanding current and emerging treatment modalities. Recommendations in this review are based on the guidelines released by the Brain Trauma Foundation in 2016 [6]. 

Each section on the pathophysiological basis is followed by a discussion of the specific treatment evolved and its current status. Treatments with multiple action mechanisms are primarily discussed in the section with a more experimental or clinical background. We also searched the clinicaltrials.gov website to evaluate any ongoing clinical trials (status: “recruiting patients”, “not yet recruiting”, or “active, not recruiting”).

## 3. Literature Review

### 3.1. Spectrum of TBI Secondary Injuries

#### 3.1.1. Mild Traumatic Brain Injury (mTBI)

mTBI, commonly known as concussion, is characterized by a range of subtle yet significant biochemical and cellular changes [7]. These changes primarily include temporary dysfunction in neuronal connectivity and metabolism [1]. Key features often involve minor axonal disruptions, alterations in ion channels leading to ionic fluxes, and imbalances in neurotransmitters [8]. While the structural integrity of the brain may largely remain intact, these molecular and cellular disturbances can result in symptoms such as headache, dizziness, and short-term memory disturbances [8]. Despite the transient nature of these symptoms, a subset of patients might experience prolonged post-concussive syndromes, wherein symptoms persist for weeks or months [8]. This phenomenon underscores the need for a nuanced understanding of mTBI’s secondary effects, particularly in the context of repeated injuries and their cumulative impact [8].

#### 3.1.2. Severe Traumatic Brain Injury

Severe TBI is marked by more profound and immediate secondary injuries that can be life-threatening and lead to long-term disabilities. The hallmark of severe TBI includes cerebral edema (swelling of the brain), increased intracranial pressure, and significant axonal injury [9]. These injuries can disrupt the blood-brain barrier, leading to inflammatory responses and oxidative stress, which further exacerbate brain damage. Secondary insults in severe TBI may also encompass hemorrhagic lesions, hypoxia, and ischemia [10], contributing to a complex cascade of neurodegeneration. The consequences of these injuries are often severe, manifesting as long-term cognitive, behavioral, and motor impairments [10]. Management of severe TBI is multifaceted, involving acute medical interventions to stabilize the patient and extensive rehabilitation programs to aid recovery and improve quality of life.

#### 3.1.3. Subdural Hematoma (SDH)

SDH occurs when blood collects between the dura mater and the arachnoid layer of the brain [11]. This accumulation of blood leads to increased intracranial pressure, which can cause significant secondary brain injury [11]. Patients with SDH often experience a progressive decline in consciousness, headache, and hemiparesis [11]. The secondary injuries in SDH can include brain herniation, ischemic injury due to compression of blood vessels, and hypoxic injury from reduced blood flow [12]. The management of SDH focuses on surgical intervention to evacuate the hematoma and relieve pressure, followed by monitoring for potential complications like rebleeding or seizures [12].

#### 3.1.4. Subarachnoid Hemorrhage (SAH)

SAH involves bleeding into the subarachnoid space, the area between the arachnoid membrane and the pia mater surrounding the brain [13]. This type of hemorrhage is often caused by the rupture of cerebral aneurysms [13]. The primary concern in SAH is the risk of rebleeding and the development of cerebral vasospasm, which can lead to ischemic injury [14]. Secondary injuries in SAH also include hydrocephalus and elevated intracranial pressure [14]. The management of SAH is complex, often requiring nimble balancing of maintaining cerebral perfusion while minimizing the risk of rebleeding and managing vasospasms, often through medication and surgical interventions [14].

#### 3.1.5. Contusions

Cerebral contusions are essentially bruises on the brain tissue, occurring typically at the site of impact or, in some cases, at a location opposite to the point of impact (coup-contrecoup injury) [15]. These injuries result from the brain striking the inner surface of the skull [16]. Contusions are characterized by localized bleeding and swelling in the brain tissue [15]. The swelling, or edema, can lead to an increase in intracranial pressure, potentially causing a reduction in cerebral blood flow and oxygen supply [17]. In severe cases, this can escalate into a herniation syndrome, where brain tissue is displaced from its normal position [17]. The management of contusions may involve monitoring for changes in neurological status and intracranial pressure, and in some cases, surgical intervention is required to alleviate pressure [17]. Rehabilitation is often necessary to address cognitive and motor deficits resulting from the injury [18].

#### 3.1.6. Diffuse Axonal Injury (DAI)

Diffuse Axonal Injury is a form of TBI that involves widespread damage to the brain’s white matter [19]. This injury occurs when the brain rapidly shifts inside the skull, typically as a result of high-velocity impacts such as those seen in car accidents or falls from significant heights [19]. The shearing forces during the impact disrupt the axons, leading to a breakdown in neuronal communication [20]. The hallmark of DAI is a disruption of the normal signaling and connectivity within the brain, which can lead to a variety of symptoms, ranging from subtle cognitive impairments to prolonged coma [20]. Diagnosis is often challenging due to the microscopic nature of the injury and may require advanced imaging techniques like Diffusion Tensor Imaging (DTI) [21]. The treatment for DAI is largely supportive, focusing on managing symptoms and preventing secondary injuries [21]. Rehabilitation plays a critical role in the recovery process, with a focus on maximizing functional independence and cognitive recovery.

#### 3.1.7. Penetrating Brain Injuries

Penetrating brain injuries occur when an object, such as a bullet or sharp instrument, breaches the skull and enters the brain tissue [22]. These injuries are particularly severe due to the direct damage to brain tissue and the high risk of infection and inflammation [22]. The trajectory and speed of the penetrating object play a significant role in the extent of the injury. Immediate concerns include bleeding, increased intracranial pressure, and the risk of secondary injuries due to brain swelling and herniation [23]. Surgical intervention is often necessary to remove foreign objects, repair damaged tissue, and reduce intracranial pressure [23]. Antibiotic therapy is critical to prevent or treat infections. Long-term rehabilitation is required to address physical, cognitive, and psychological impacts [24]. Due to the complex nature of these injuries, outcomes can vary widely, with some individuals experiencing significant long-term disabilities.

### 3.2. Pathophysiology of Secondary Brain Injury

Secondary traumatic brain injury is the neurological damage that evolves due to the primary injury (Figure 2). Within the first few hours following the injury, a series of events may trigger a cascade of secondary metabolic, inflammatory, and ischemic insults that can exacerbate the primary neuronal injury. These events are associated with the activation of genes that lead to the transcription of proteins and enzymes involved in the release of excitatory amino acids, the production of free radicals and inflammatory cytokines, alterations in ion flux across membranes, upregulation of neuroprotective cascades, and the induction of programmed cell death [25].

Alterations in cellular metabolism and signaling pathways (Figure 3) can lead to the depletion of energy reserves. As a result, neuronal cells may no longer be able to maintain depolarization of their membranes, causing alterations in ion fluxes and osmotic swelling of the cells. Therefore, following TBI, it is crucial to recognize, prevent, and treat secondary injury to ensure neuronal survival.

### 3.3. Sequelae from the Primary Impact

#### 3.3.1. Cellular Events

##### Excitotoxicity and Calcium

Severe TBI causes an abrupt increase in extracellular excitatory amino acids such as glutamate and aspartate [2]. Evidence shows a 40% decline in the expression of astrocytic sodium-dependent glutamate transporters GLAST (EAAT1) and GLT-1 (EAAT2) within 24 h following TBI, leading to a significant decrease in the resorption of glutamate [2]. These excitatory amino acids activate N-methyl-D-Aspartate (NMDA) and non-NMDA receptors, leading to cell membrane depolarization and the influx of sodium (Na^2+^), potassium (K^+^), and calcium (Ca^2+^). High intracellular calcium concentrations activate phospholipase and calpain enzymes that alter membrane and cytoskeletal integrity, eventually resulting in neuronal cell damage and death. 

Magnesium (Mg^2+^) can regulate excitotoxic processes by blocking NMDA receptors and calcium channels. However, Mg^2+^ concentration decreases after TBI and persists for at least four days after the impact, and its deficiency has been associated with poor neurological outcomes [26]; increasing extracellular magnesium concentration improves the recovery of hippocampal neuronal high-energy phosphates and accelerates regional cerebral flow to the ischemic brain area [27]. 

Cortical spreading depression (CSD) is a slowly propagating wave of depolarization of neurons and glial cells, followed by a sustained suppression of spontaneous neuronal activity, accompanied by complex and variable changes in vascular caliber, blood flow and energy metabolism [28]. Excitotoxicity is complicated when cortical spreading depression (CSD) occurs [3], a depolarization wave in cerebral gray matter that propagates across the brain [28]. It can break down ion homeostasis, facilitating the release of excitatory amino acids. CSD is associated with poor outcomes in TBI patients [28]. In humans, microdialysis studies demonstrate an increased lactate-pyruvate ratio without consistent evidence of ischemia, and this finding is strongly correlated with outcome [29]. CSD and/or mitochondrial dysfunction may underlie this marker of disturbed metabolism since they are associated with lactate accumulation [25,30].

##### Specific Treatment

Until this point, statins have been proven to have the potential to protect cultured neurons from excitotoxic death caused by the glutamate receptor agonist NMDA [31]. Interestingly, when evaluating TBI patients with prolonged disorders of consciousness, the use of amantadine, an assumed NMDA antagonist and dopamine agonist, accelerates functional recovery [32]

Multiple clinical trials in acute TBI targeting glutamate and specifically its NMDA receptor have shown contradictory results to demonstrate any beneficial effect [33,34,35], as well as treatment with nimodipine (a calcium channel blocker) [36]. Some authors argue that blocking synaptic transmission mediated by NMDA receptors hinders neuronal survival [34].

Despite promising experimental studies with the use of intravenous magnesium sulfate, its clinical efficiency has yet to be proven magnesium sulfate therapy is effective in the treatment of patients with TBI [37,38].

Regarding CSD, clinical management can focus on controlling factors that increase its incidence and duration, such as systemic hypotension, pyrexia, hypoxia, and low plasma glucose [28].

##### Free Radicals and Oxidative Stress

Free radicals are by-products of energy metabolism in cells and play a role in vascular tone and immune function. Signaling pathways can increase free radical formation after TBI. Calcium activates pathways that release free radicals from mitochondria and increase nitric oxide (NO) production through inducible NO synthase (iNOS) [39]. Nicotinamide adenine dinucleotide (NADPH) oxidase produces reactive oxygen species (ROS) within the first hour after trauma [40]. ROS increases affinity to NO, forming peroxynitrites (ONOO-) [40], which destroy the cytoskeleton, cell membranes, and DNA [41]. NO inhibits cytochrome c oxidase, causing mitochondrial disruption and cell death [42].

NO formed by vascular endothelium (endothelial nitric oxide synthase—eNOS) exerts beneficial effects after TBI, including vasodilation and increased cerebral blood flow [42]. Peroxynitrites are the leading cause of oxidative stress, as indicated by increased nitrotyrosine in TBI patients with poor outcomes [43]. ONOO- inhibits potassium channels, increasing vascular tone and impairing vascular reactivity [42]. Iron compounds from hemoglobin degradation form reactive free radical oxidants, altering synaptic function and contributing to posttraumatic seizures [44].

Specific treatment

Citicoline has multiple neuroprotective mechanisms, including inhibition of oxidative stress and apoptotic pathways [45]. However, a large randomized controlled trial (RCT) showed no benefit in TBI patients [46]. 

Statins up-regulate eNOS expression and inhibit inducible NO, interleukin1 β (IL1 β), and tumor necrosis factor α (TNF α) [46]. They reduce post-traumatic hypoperfusion and rebound hyperemia [46], protect neurons from excitotoxic death [31], and may reduce cerebral edema and intracranial hypertension [47]. Susanto et al. demonstrated that compared to nonusers, either simvastatin 40 mg, atorvastatin 20 mg, or rosuvastatin 20 mg for ten days reduced mortality risk in TBI individuals. In contrast, statin discontinuation was associated with increased mortality [47].

##### Inflammatory Mediators and Cascades

Traumatic injuries cause disturbances in the normal cellular functioning of the brain due to the impact of direct, rotational, and shear forces [48]. Axonal injury leads to localized swelling, which hampers the transmission of signals. Traumatic injuries are also connected to alterations in cerebral blood flow, causing an initial decrease in blood circulation and subsequent unresponsive vasodilation, believed to be caused by the release of nitric oxide in the affected tissue [49]. 

Following a focal injury, the first component of the neuronal structure to be affected at the cellular level is the axonal membranes, given the rotational and direct forces. This axonal damage leads to the release of potassium from the intracellular environment. After membrane depolarization, calcium entry through voltage-dependent channels promotes the release of excitatory amino acids and neurotransmitters. At this point, potassium and calcium freely move between the intra- and extracellular spaces, disrupting intracellular homeostasis [48]. 

The impairment of intracellular functions leads the neuronal tissue into anaerobiosis [50]. Lactate levels rise, further contributing to local damage to the blood-brain barrier and cell death. This process can occur for 4 to 6 h [51].

In brief, the disruption of cellular membranes resulting from the primary mechanical insult or secondary injury triggers the release of damage-associated molecular patterns (DAMPs). This prompts the rapid upregulation of tumor necrosis factor (TNF), Interleukin 6 (IL-6), and Interleukin 1β (IL-1β) by local glial cells and infiltrating immune cells, acting as early mediators that drive the inflammatory response following traumatic injury [52].

In the context of TBI, microglia, play a crucial role. Traditionally, microglia have been classified into M1 (pro-inflammatory) and M2 (anti-inflammatory). This classification is based on their response to different micro-environmental disturbances 1, 2.

DAMPs release by injured neurons and proinflammatory and oxidative mediators from infiltrating immune cells leads to microglial cells polarization towards an M1-like phenotype [53]. The expression of proinflammatory factors (Table 1) such as IL-1β, TNF, IL-6, Nitric oxide synthase 2 (NOS2), Interleukin 12p40 (IL-12p40), and NADPH Oxidase 2 (NOX2) characterizes M1-like cells.

In response to anti-inflammatory and neurotrophic signals, microglia and macrophages can shift towards an M2-like phenotype [70]. M2-like cells express proteins such as CD206, CD163, arginase-1, FCγR, Ym1, IL-10, and TGFβ. Molecular pathways involved in regulating M2-like phenotypic transitions include STAT6/3, IRF-4/7, NF-κB p50/p50, Nrf2, and miR-124. M2-like microglia and macrophages release anti-inflammatory and trophic factors, promoting the resolution of inflammation [71,72]. Microglia and macrophages possess remarkable plasticity and can transition between M1-like and M2-like phenotypes. Mixed phenotypes are present in the acute phase following TBI, eventually transitioning to an M1-like dominant phenotype in the phase [72,73].

However, this binary classification has been interpreted somewhat arbitrarily, as there is a continuum of different intermediate phenotypes between M1 and M2, and microglia can transition from one phenotype to another, contributing to neurodegeneration, as seen in Alzheimer’s Disease animal models [72].

More recent studies have proposed a different categorization of microglia as resting, activated, or “disease-associated microglia” (DAM). DAM was initially characterized in mouse models of Alzheimer’s disease [71]. In DAM, homeostatic microglial markers are downregulated, and other genes, including Trem2, Apoe, Itgax, Clec7a, Axl, and Lpl, among others, are upregulated, playing a relevant job in the classification of microglia state. In this context, it has been proposed that rather than an initial M1 phenotype, in secondary TBI injury, the pathway to neurodegeneration follows an initial, activated state, and a progression DAM [71,72,73].

1.Treatments based on the pathway level

The development of innovative anti-inflammatory drugs for managing TBI is facilitated by targeting various signaling pathways such as NF-κB, MAPKs, JAK/STAT, PI3K/Akt/mTOR, and TGF-β1 [74]. Once inflammatory mediators are released, immune response and glial cells are recruited. Microglial cells form the first line of differentiation between the intact and injured tissue [73,75]. Microglial cells release oxidation metabolites and pro-inflammatory reactants and cytokines such as interferon-gamma, interleukins, and tumor necrosis factor-alpha, especially in the latter. All of this leads to the stimulation of astrocytes for the formation of glial scars at the sites of trauma [27].

Nuclear factor-kappa B (NF-κB) is a transcription factor that regulates the synthesis of inflammatory molecules, pro-inflammatory cytokines, and chemokines [76]. In glial cells, NF-κB promotes inflammation, while in neurons, it is associated with neuroplasticity and neuronal development. Inhibiting this factor could reduce apoptosis and secondary inflammation in TBI [77,78].

In TBI, the Janus Kinase/Signal Transducer and Activator of Transcription (JAK/STAT) pathway decreases its expression, leading to increased cell death. In a study, recombinant erythropoietin was administered to cortical cells of rats with prior TBI, resulting in an increase in JAK and STAT and a reduction in apoptosis [79].

The MAPK (Mitogen-Activated Protein Kinase) pathway plays a crucial role in cell differentiation, proliferation, and survival. The path consists of c-Jun NH (2)-terminal kinase (JNK), extracellular signal-regulated protein kinase (ERK), and p38 [80]. Several studies have indicated that activating the p38 and JNK pathways contributes to increased neuronal damage in spinal cord injury and cerebral ischemia cases, activating a complex cascade in the mitochondria of brain cells and leading to apoptosis [81].

The dual nature of inflammation has been illustrated in experimental models exploring the involvement of TNF and inducible nitric oxide synthase (iNOS) following TBI [82]. TNF has been associated with brain edema, BBB disruption, and leukocyte recruitment [83]. Surprisingly, mice lacking TNF exhibited motor function impairment and larger lesions four weeks after the injury despite showing early neuroprotection [84]. Similarly, while TBI increased iNOS expression in the brain, which has multiple proinflammatory and neurotoxic effects, the genetic or chemical blockade of iNOS worsened spatial memory two to three weeks after the injury [85,86,87]. Cell death through programmed necrosis, such as necroptosis triggered by TNF-mediated receptor-interacting protein (RIP) kinase activation, can initiate a detrimental cycle: necrosis leads to further membrane disruption, promoting the release of DAMPs, which, in turn, exacerbates necrosis and amplifies inflammation [85,86,87]. 

2.Other neuroinflammatory components

In response to TBI and glutamate toxicity, endogenous neuroprotectant adenosine levels are produced due to adenosine triphosphate and mRNA breakdown [88,89]. Following the ATP production and the mRNA, the activation of adenosine receptor A1 after TBI has been found to have anti-excitotoxic and anti-inflammatory effects in mice. However, it is essential to note that systemic administration of adenosine to patients can lead to bradycardia and hypotension [88,90]. 

Other components, including free radicals, lipid peroxidation, and direct impact, trigger the release of inflammatory mediators such as cytokines, chemokines, and complement. Cytokines are signaling molecules produced by immune system cells and brain cells, including microglia, astrocytes, and neurons. They act synergistically in cascades. After TBI, tumor necrosis factor α and interleukin1 β trigger the formation of peptides, ROS, and nitrogen species [91]. These mediators perpetuate secondary brain injury by activating arachidonic acid and coagulation cascades and disrupting BBB [92].

Blocking cytokines ameliorates pathological consequences in TBI models [93,94]. However, some cytokines modulate trophic responses and restorative functions [95] (Table 1). They can be divided into pro- and anti-inflammatory molecules.

3.Specific treatment

The activation of inflammasomes, which leads to the release of IL-1β, can be inhibited by targeting IL-1 receptors, which can inhibit the invasion of circulating immune cells into the CNS [96]. 

Treatments like intravenous immunoglobulin can inhibit the priming of T cells from entering the CNS [97]. Furthermore, alterations in the gut microbiome may influence the balance between proinflammatory and anti-inflammatory T lymphocytes [98]. Impaired glymphatic clearance after TBI may hinder the removal of proinflammatory mediators, and ongoing investigations aim to enhance glymphatic flow, with increased clearance observed during sleep [99].

Nonsteroidal anti-inflammatory drugs (NSAIDs) produce an inhibition of COX that significantly reduces the levels of IL-1β and hinders the synthesis of IL-6 by modulating the pathways involved in producing vasodilator prostaglandins [100]. Another critical aspect of NSAIDs is their ability to stimulate the proliferator-activated receptor (PPAR), which elicits transcriptional regulatory effects that reduce the levels of various proinflammatory substances and decrease microglial activity. Additionally, several NSAIDs exhibit antioxidant properties and can inhibit the activation of NF-kB [101].

##### Mitochondrial Dysfunction

Impaired mitochondrial function and the generation of reactive oxygen species (ROS) are recognized consequences of direct and indirect traumatic brain injuries (TBIs) that contribute to the initiation of neuroinflammation [102]. Experimental evidence has shown that the movement of the phospholipid cardiolipin from the inner to the outer mitochondrial membrane occurs following TBI, marking damaged mitochondria for selective removal through mitophagy [103]. These mitochondrial signals of danger prompt local and systemic responses by interacting with specific receptors on immune cells [104]. For instance, mitochondrial DNA interacts with Toll-like receptor 9 (TLR9) on dendritic cells, while N-formyl peptides bind to formyl peptide receptor 1 on neutrophils [105].

After traumatic brain injury (TBI), energy storage may be depleted. Mitochondria play a crucial role in producing ATP. Neuronal cells maintain ATP levels through phosphocreatine deposits, glycolysis, and oxidative phosphorylation via the Krebs cycle and respiratory chain. The latter is the most efficient and occurs entirely in mitochondria. Lactate, a result of pyruvate oxidation, is the main substrate for energy production [105]. Impairments in the electron transport system (ETS) increase reactive oxygen species (ROS) formation and decrease energy production [106]. 

Mitochondrial dysfunction in severe head injury is supported by inadequate oxidative mitochondrial metabolism, mitochondrial swelling, and decreased ATP production [107,108]. Dysfunction begins early and may persist for days. Lipid peroxyl radicals contribute to secondary mitochondrial dysfunction [109]. Activation of the mitochondrial permeability transition pore (mPTP) determines cell survival after TBI [110]. Excess Ca^2+^ within mitochondria can cause mPTP opening, leading to mitochondrial edema and decreased oxidative metabolism [110]. mPTP opening allows extrusion of mitochondrial Ca^2+^ and activates harmful calcium-dependent proteases such as calpain [111]. Loss of cytochrome c through mPTP activates cellular apoptosis via caspase 9 [109,111]. Therapeutic interventions targeting mitochondria may delay or prevent secondary cascades leading to cell death and neurobehavioral disability [111].

Specific treatment

Cyclosporine A (CsA) stabilizes the mitochondrial permeability transition pore (mPTP) and has demonstrated neuroprotection in pre-clinical TBI studies [112]. Sullivan et al. showed on animal models that animals receiving CsA demonstrated a reduction in the lesion volume, even up to 74% [113]. Another mechanism that has been explored is the reduction of brain metabolic activity through hypothermia [113,114]. However, a recent meta-analysis conducted by Chen et al., which included 23 trials involving 2796 patients, demonstrated that therapeutic hypothermia did not reduce, but surprisingly, can increase the mortality rate of patients with TBI in some high-quality studies. However, the therapy can benefit patients with demonstrated increased intracranial pressure, not as a prophylactic therapy but as a therapy within the first 24 h [114]. 

Finally, barbiturates suppress cerebral metabolism and reduce cerebral blood volume and intracranial pressure (ICP) [115]. In a recent multicenter European trial, high-dose barbiturate treatment caused a decrease of 69% in ICP; however, this effect was also accompanied by hemodynamic instability, leading to more extended periods of mean arterial pressure <70 mmHg despite the use of vasopressors [115].

##### Cell Death

Cell death after TBI can occur through necrosis or apoptosis. Both can occur in regions remote from the impact site within days and weeks after trauma. Necrosis occurs in response to severe mechanical or ischemic/hypoxic tissue damage, excessive excitatory amino acid neurotransmitter release, and metabolic failure [116]. Caspases and calpain are important mediators of programmed cell death (apoptosis), with calpain activation more associated with necrosis [116].

Caspases are activated through the extrinsic pathway initiated by cell surface death receptor ligation (receptor-linked caspase-8 pathway) and the intrinsic pathway arising from mitochondria (mitochondrial caspase-9 pathway) [116]. Caspase 3 compromises membrane permeability to Ca^2+^, leading to elevated intracellular Ca^2+^ levels. Caspase 3 also degrades calpastatin, facilitating calpain activation [117]. Endogenous inhibitors, such as the inhibitors of the apoptosis family, modulate caspase activity within these pathways [117].

The activation of caspase-3 occurred via extrinsic and intrinsic apoptotic pathways after TBI has been documented, and this response is also mirrored in the retina [118]. However, experimental studies suggest that when activation is sustained, calpain is more important than caspase-3 in mediating cell death after TBI [118,119]. The anti-apoptotic modulator B-cell lymphoma (Bcl-2) inhibits mPTP, preserving mitochondrial homeostasis and preventing mitochondrial Ca^2+^ leak and programmed cell death [120]. Several in vivo studies with transgenic mice have shown promising results, at least with a reduced deficit in mice overexpressing the BCL-2 [121]. In clinical trials, a literature review published by Deng et al. showed reduced mortality and better outcomes in the Glasgow Coma Score (GOS) in the patients with increased Bcl-2 in the peritraumatic tissue, highlighting the importance of the peptide as a potential biomarker and therapeutic target [122].

Specific treatment

Erythropoietin (Epo) and its receptor (Epor) are expressed throughout the central nervous system [123]. Epo modulates caspase 1, caspase 3, and caspase 8-like activities, maintaining genomic DNA integrity and preventing acute cellular injury and microglial activation [124]. Epo also modulates mitochondrial membrane permeability and cytochrome c release, reduces cerebral vasospasm, and improves cerebral blood flow when given early post-injury [124]. 

#### 3.3.2. BBB Disruption and Neutrophil Invasion

BBB regulates the exchange of substances between plasma and brain interstitium [125]. This barrier is formed by brain endothelial cells connected by tight junctions, with mechanical support from astrocytes critical for normal function [125]. Ion homeostasis and uptake of small molecules are conducted via specific endothelial membrane channels and solute carriers. Larger peptides and proteins are transported by endo- or transcytosis pathways within caveolae and clathrin-coated microvesicles. Paracellular diffusion is restricted by tight junctions between adjacent endothelial cells [126]. BBB disruption after injury is typically biphasic, with an immediate phase of hyperpermeability, temporary restoration of BBB function, and a delayed opening period [126]. BBB disruption leads to excess interstitial water (vasogenic edema) accumulation, contributing to cerebral swelling, brain shift, and herniation [125].

As previously mentioned, the BBB tight junction is based on the interaction between endothelial cells and astrocytes. Some studies have shown that the mechanical damage to astrocytes initiates oxidative-stress-mediated edema, altering astrocyte ionic gradients and increasing BBB permeability [125,127]. ROS increase brain endothelium permeability and promote post-traumatic invasion of inflammatory cells by upregulating endothelial expression of cell adhesion molecules such as intercellular adhesion molecule-1 (ICAM1) [127]. Degradation of membrane proteins and increased permeability contribute to BBB opening, vasogenic edema formation, and increased intracranial pressure [128].

Inflammatory cells provide the primary source of matrix metalloproteinase (MMP) activity [129]. MMPs promote cell death, including apoptosis [129]. Other inflammatory mediators in BBB opening include substance P, kinins, and bradykinins [128]. Particularly, substance P and its receptor, neurokinin 1 (NK1R), have been promising [130]. MMPs are released in ischemic brain injury and contribute to BBB disruption by degrading basal lamina components and tight junctions [129]. Later, MMPs are involved in tissue remodeling and neurovascular recovery. Vascular endothelial growth factor (VEGF) alters BBB permeability by changing the distribution and downregulating the expression of tight junction proteins [129].

##### Specific Treatment

Substance P antagonists (SP, NK1 receptor antagonist) show promising results in limiting BBB opening and edema after TBI [130]. Vink et al., in their review article, showed that NK1 antagonists can reduce posttraumatic ICP and improve brain oxygenation after TBI [90]. In vitro studies have shown the potential for the NK1-R antagonist to reverse the compromise, integrity, and function of the BBB [131].

Another alternative is Hyperbaric Oxygen Therapy (HBOT). HBOT reduces MMP-9 expression and inhibits neutrophilic infiltration. It may also counter capillary vasodilation within hypoxic tissues [132]. Hadanny et al. showed in their review that a search from 1969 to 2023 showed that HBOT should be recommended in acute moderate-severe TBI, specifically for patients suffering from prolonged post-concussion syndrome who have precise evidence of metabolic dysfunctional brain regions. However, further studies are needed to evaluate outcomes and determine the optimal treatment protocols [132] (Table 2).

## 4. Conclusions

Managing TBI patients includes specialized prehospital care, intensive clinical care, and long-term rehabilitation. However, neuroprotective agents to limit secondary injury or enhance repair lack clinical effectiveness. The complexity of TBI pathophysiology may reflect the difficulty of translating preclinical benefits into clinical practice. The current goal is to follow and identify the sequence of events of secondary lesions to avoid further neuronal damage. Knowledge of these concepts, the development of more efficient clinical trial designs, and the possibility of combination therapies may change the course of treatment in the acute phase in the near future.

## Figures and Tables

**Figure 1 biomedicines-12-00520-f001:**
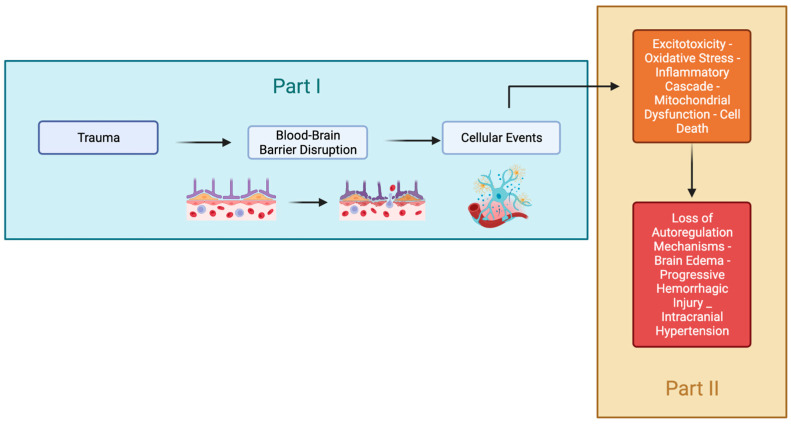
The summarized pathophysiology of TBI. This review will be separated into two parts. Part I will focus on the cellular and Brain blood barrier (BBB) that evolve following the injury. Part II will focus on the consequence of these molecular mechanisms and its repair.

**Figure 2 biomedicines-12-00520-f002:**
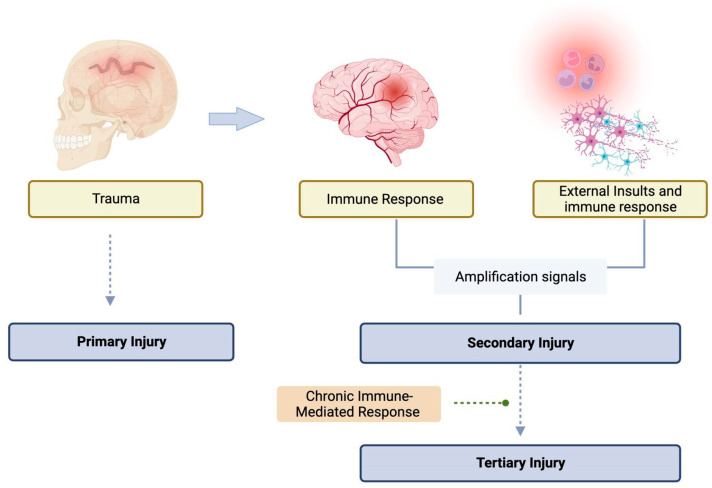
Schematic view of the primary, secondary, and tertiary injuries after TBI. Solid arrows indicate an association, and dashed arrows indicate the following stage (leads to).

**Figure 3 biomedicines-12-00520-f003:**
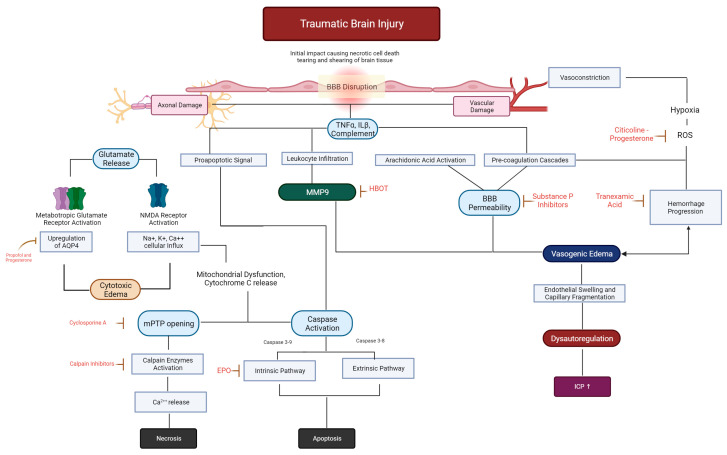
Schematic evolution of the major mechanisms involved in the secondary injury of TBI. The sequence of events can occur simultaneously and interact to exacerbate injury and initiate neuronal and vascular repair. Potential drugs are placed in red-doted arrows directed to their leading site of action. Black solid lines indicate association, while red continuous indicate an inhibitor of the pathway.

**Table 1 biomedicines-12-00520-t001:** Main cytokines ^1^ involved in traumatic brain injury.

	Main Effects in TBI	Highlights
IL1β	Detrimental	Raised CSF IL1β correlates with both raised ICP and poorer outcome. The balance between members of the IL1 cytokine family, in particular between IL1β and its endogenous inhibitor IL1ra, is an important determinant of the degree of inflammatory response, rather than the absolute concentration of IL1β. In TBI patients, high microdialysate IL1ra/IL1b ratio is associated to favourable outcomes [54,55,56].
TNFα	Detrimental in acute phase and beneficial during the healing process	Upregulated in the injured brain early after trauma, reaching a peak within a few hours following the initial injury. This cytokine triggers the apoptotic cascade but also, pathways resulting in activation of pro-survival genes [57,58,59].
IL6	Beneficial	A trophic factor that is upregulated in the CNS after injury and promotes neuronal survival and brain repair through astroglia and vascular remodelling. Following TBI, its concentration rises dramatically [60,61].
Complement	Detrimental	The complement system significantly triggers inflammation in TBI, increasing BBB permeability and inducing cytokines, chemokines, and adhesion molecules. Key products like C3a and C5a enhance vascular permeability and inflammation. The cascade forms the membrane attack complex (MAC), affecting both non-nucleated and nucleated cells. C5a, a major pro-inflammatory component, amplifies cytokine production and leukocyte adhesion, leading to more inflammation. This system also contributes to apoptosis and cell death in TBI [62,63,64].
Angiopoietins	Beneficial (Ang1) and detrimental (Ang2)	They are family of growth factors that are important in regulating angiogenesis and vascular permeability, and also have been implicated in BBB disruption. TBIs models show acute decrease in Angiopoetin-1 expression and concomitant increase in Angiopoetin-2 which is associated with endothelial apoptosis and BBB permeability [65,66,67,68,69].

^1^ IL1β: interleukin1β; TNFα: tumor necrosis factor α; IL6: interleukin6.

**Table 2 biomedicines-12-00520-t002:** Summary of Potential Therapeutic Targets and Drugs for TBI.

Pathophysiology	Target/Drug	Clinical/Preclinical Evidence	Key Findings/Comments
Excitotoxicity and Calcium	Statins	Cultured neurons protection from NMDA-induced death [31]	Potential protection, contradictory clinical results
	Amantadine	Accelerates functional recovery in TBI patients [32]	Beneficial for prolonged disorders of consciousness
	Nimodipine (Calcium channel blocker)	Contradictory results in clinical trials [36]	Some argue hindrance of synaptic transmission
	Magnesium Sulfate	Promising in experimental studies, clinical efficacy TBD [37,38]	Potential effectiveness in TBI treatment
Free Radicals and Oxidative Stress	Citicoline	Multiple neuroprotective mechanisms [45], no benefit in RCT [46]	Inhibition of oxidative stress and apoptotic pathways
	Statins	Upregulate eNOS, reduce hypoperfusion, protect neurons [46]	Reduction in mortality risk in TBI individuals
Inflammatory Mediators and Cascades	Innovative Anti-inflammatory Drugs	Targeting NF-κB, MAPKs, JAK/STAT, PI3K/Akt/mTOR, TGF-β1 [74]	Potential for reducing apoptosis and inflammation
	Recombinant Erythropoietin	Increases JAK and STAT, reduces apoptosis in TBI [79]	Potential for neuroprotection
	MAPK Pathway (p38, JNK)	Activation contributes to increased neuronal damage [80]	Inhibition may reduce neuronal damage
	TNF and iNOS	Dual nature of inflammation, complex effects [82]	TNF associated with brain edema, iNOS inhibition worsens spatial memory [85,86,87]
Other Neuroinflammatory Components	IL-1 Receptor Inhibitors	Inhibit inflammasome activation, reduce IL-1β release [96]	Potential to limit neuroinflammation
	Intravenous Immunoglobulin	Inhibit priming of T cells from entering CNS [97]	Potential inhibition of T cell infiltration
	NSAIDs	Inhibition of COX, reduction in IL-1β, anti-inflammatory [100]	Modulation of proinflammatory pathways
Mitochondrial Dysfunction	Cyclosporine A (CsA)	Stabilizes mPTP, neuroprotection in preclinical TBI [102]	Reduction in lesion volume in animal models
	Therapeutic Hypothermia	Mixed results in meta-analysis, potential benefits in specific cases [103]	Not universally beneficial, effective in increased intracranial pressure
	Barbiturates	Suppress cerebral metabolism, reduce ICP [114]	Significant ICP decrease but accompanied by instability
Cell Death	Erythropoietin (Epo)	Modulates caspase activities, mitochondrial function [124]	Potential for neuroprotection, modulation of cell death
BBB Disruption and Neutrophil Invasion	Substance P Antagonists	Limit BBB opening, edema after TBI [130]	Promising results, potential BBB protection
	Hyperbaric Oxygen Therapy (HBOT)	Reduces MMP-9, inhibits neutrophilic infiltration [132]	Potential benefits in acute moderate-severe TBI

## Data Availability

Not applicable.

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
