# Peer review of "Pathophysiology-Based Management of Secondary Injuries and Insults in TBI"

_biomedicines, 2024, doi:10.3390/biomedicines12030520_

Round 1

Reviewer 1 Report

Comments and Suggestions for Authors

Filho et al present a review of molecular pathways involved in secondary injuries in TBI.

In general, this review is confusing and difficult to follow. It attempts to include a large portion of molecular biology into a very fragmented and condensed review of the literature. After reading this review it is not clear what the message is, except of the complex nature of the problem.

1.     The authors should outline first what are the main clinical secondary injuries by severity and type of TBI. For example, secondary injuries in SDH are different than SAH and mTBI is different than severe TBI. Age plays a significant factor in the subject as well.

2.     In 2. Genes don’t change to release amino acids…they get activated and transcribe information.

3.     Fig 3 is too busy and frankly difficult to visualize.

4.     In 3.1 second paragraph what is (Health)? Last line , no explanation of what “microdialysis studies” are. In the first paragraph, last line “will be explained later (2)” needs to be omitted.

5.     In  3.1.3 3rd paragraph, first sentence it says (cite)- no citation is present.

6.     Polarization to M1 phenotypes is in microglial cells which is not mentioned in the 5th paragraph (line 176).

7.     Table 1 should have appropriate references. IL6 is not a VEGF. Complements- which ones?

8.     No definition of what is “chronic TBI” on page 5.

9.     What is DAI, on page 5?

10.  No definition of what CSD refers to.

11.  On page 9 “specific treatment” paragraph 2 and three are repeated, verbatim in 3.1.4.

This review is a collection of statements/facts that appear to be copied with disrupted flow. Better and more illustrations could perhaps improve it.

Comments on the Quality of English Language

minor editing needed.

Reviewer 2 Report

Comments and Suggestions for Authors

The article attempted to review pathophysiology based TBI management using cellular and molecular mechanisms and their related treatment, which is a novel thought; however, if the article takes TBI of a simple blow such as blast wave injury to the head to a closed head and penetrating injury mechanisms, including mild to severe injuries, into account. 

Major and minor comments are below: 

Page 1, Line 30:  Why the secondary injury within the brackets?

Page 2, Lines 53-55:   In the figure legend, use either TBI instead of traumatic brain injury or expand as a blood-brain barrier instead of BBB. Make it uniform while writing.

Page 4, Figure 3: The figure needs improvement in quality and does not represent all the parts of cellular events like free radicals and their treatment, inflammatory mediators and cascades and their treatment, etc.

Page 5, Line 117: What is DAI? It's the first time used.

Page 5, Line 122:  Use only Free Radicals and Oxidative Stress to make it uniform, as written in 3.1.1

Page 5, Line 124: Use only TBI, as mentioned earlier.

Page 5, Lines 141-144: Figure 3 does not represent free radicals and Oxidative stress and does not mention Citicoline/Statin treatments.

Page 6, Line 174: Use only DAMPs as the term already introduced in the previous paragraph.

Page 6, Lines 171-172: IL-6 and IL-1β -The terms first used and expanded like TNF.

Page 6, Line 177: NOS2, IL-12p40, and NOX2- The terms are used first and expanded like TNF.

Page 7, Table 1: Complement – not appropriately explained with proper examples.

Page 8, Line 196: Use only TBI as the earlier mentioned term.

Page 8, Line 228:  What is RIP kinase? It's the first time used.

Page 9, Line 230:  Use only DAMPs as the term already introduced previously.

Page 9, Line 244: Use only TBI as a previously introduced term.

Page 9, Line 245: Use only ROS as a previously introduced term.

Page 9, Line 247:  Use BBB instead of the blood-brain barrier, as the term was already introduced previously.

Page 9, Lines 252-268: Major problem in writing The Specific treatment part is repeated on Page 9, Lines 270 -286 with 3.1.4. Mitochondrial Dysfunction in Traumatic Brain Injury.

Page 9, Line 270: Use only Mitochondrial Dysfunction to make uniform as written in 3.1.1.

Page 9 Lines 270-286: Repetition of the Page 9 Lines 252-268: Specific treatment part.

Page 10, Line 308: Use only TBI as a previously introduced term.

Page 10, Line 321:  Use only TBI as the term already introduced previously.

Page 10, Line 325:  Use only mPTP as the term already introduced previously.

Page 10, Line 343: Use only BBB as previously introduced.

Page 10, Line 357: Use only ROS as a previously introduced term.

Page 12, Line 388: Use only TBI as a previously introduced term.

Reviewer 3 Report

Comments and Suggestions for Authors

The authors have summarized some of the common secondary injuries after TBI. They also reviewed some relevant management/treatment approaches that have been reported (clinically and pre-clinically). This is an important topic, as currently effective treatment for TBI is still lacking.

There are some major comments to the article:

1. Pathophysiology of secondary injury after TBI

The major mechanisms discussed in this article include excitotoxicity (ionic fluxes), neuro-inflammation and oxidative stress, and BBB disruption.

However, there are also some other major mechanisms (such as energy metabolism changes and axonal injury) which are not fully discussed in the review. The reviewer should clarify that this review only focuses on the aforementioned mechanisms.

2. Literature search

The authors mentioned the database they used for this review. However, they should also describe other parameters, such as the search key words, date of search, and number of hits returned.

3. P.8 Line 184-193

The authors discussed M1/M2 microglia classification. However, more recent papers are finding this classification system arbitrary. Instead more recent studies categorize microglia as resting, activated, or "disease-associated microglia" (DAM). The DAM markers (some markers include Ctsb/d/s, Trem2, Tyrobp, Clec7a, etc) are thought to be more relevant to the function and status of microglia.

5. The authors have reviewed and cited a number of clinical/preclinical articles that have different targets/drugs for different pathophysiology-based management. They should consider to tabulate and summarize these articles with a table, and include relevant information (e.g. TBI severity, sample size, or type of preclinical TBI models). This may help the readers to understand what management works (or does not work) for what type of TBI.

Minor comments/Typo:

5a.  In the review, a lot of the cations did not have the proper superscript (Ca2+, K+, etc)

b. In Fig 3. "Phospholipase" should be used instead of "Fospholypase"

c. P.6, Line 166 - one citation is missing. "(cite)"

d. Table 1. The findings summarized here should be cited as well.

e. P.8 Line 205

NFkB is a transcription factor that regulates the synthesis of inflammatory molecules, rather than "NFkB synthesizes inflammatory molecules".

Round 2

Reviewer 1 Report

Comments and Suggestions for Authors

The authors made substantial changes that improved the readability and quality of this review.

Reviewer 2 Report

Comments and Suggestions for Authors

The authors responded to the comments.